# Growth Hormone (GH) Crosses the Blood–Brain Barrier (BBB) and Induces Neuroprotective Effects in the Embryonic Chicken Cerebellum after a Hypoxic Injury

**DOI:** 10.3390/ijms231911546

**Published:** 2022-09-30

**Authors:** Rosario Baltazar-Lara, Janeth Mora Zenil, Martha Carranza, José Ávila-Mendoza, Carlos G. Martínez-Moreno, Carlos Arámburo, Maricela Luna

**Affiliations:** Departamento de Neurobiología Celular y Molecular, Instituto de Neurobiología, Campus Juriquilla, Universidad Nacional Autónoma de Mexico, Querétaro 76230, Mexico

**Keywords:** hypoxia, growth hormone, cerebellum, blood–brain barrier, neuroprotection

## Abstract

Several motor, sensory, cognitive, and behavioral dysfunctions are associated with neural lesions occurring after a hypoxic injury (HI) in preterm infants. Growth hormone (GH) expression is upregulated in several brain areas when exposed to HI conditions, suggesting actions as a local neurotrophic factor. It is known that GH, either exogenous and/or locally expressed, exerts neuroprotective and regenerative actions in cerebellar neurons in response to HI. However, it is still controversial whether GH can cross the blood–brain barrier (BBB), and if its effects are exerted directly or if they are mediated by other neurotrophic factors. Here, we found that in ovo microinjection of Cy3-labeled chicken GH resulted in a wide distribution of fluorescence within several brain areas in the chicken embryo (choroid plexus, cortex, hypothalamus, periventricular areas, hippocampus, and cerebellum) in both normoxic and hypoxic conditions. In the cerebellum, Cy3-GH and GH receptor (GHR) co-localized in the granular and Purkinje layers and in deep cerebellar nuclei under hypoxic conditions, suggesting direct actions. Histological analysis showed that hypoxia provoked a significant modification in the size and organization of cerebellar layers; however, GH administration restored the width of external granular layer (EGL) and molecular layer (ML) and improved the Purkinje and granular neurons survival. Additionally, GH treatment provoked a significant reduction in apoptosis and lipoperoxidation; decreased the mRNA expression of the inflammatory mediators (TNFα, IL-6, IL-1β, and iNOS); and upregulated the expression of several neurotrophic factors (IGF-1, VEGF, and BDNF). Interestingly, we also found an upregulation of cerebellar GH and GHR mRNA expression, which suggests the existence of an endogenous protective mechanism in response to hypoxia. Overall, the results demonstrate that, in the chicken embryo exposed to hypoxia, GH crosses the BBB and reaches the cerebellum, where it exerts antiapoptotic, antioxidative, anti-inflammatory, neuroprotective, and neuroregenerative actions.

## 1. Introduction

Perinatal asphyxia is an important cause of mortality and morbidity in preterm and term infants [1]. Hypoxia can lead to a constellation of long-term neurodevelopmental deficits, such as motor impairment, sensory disorders, and cognitive and behavioral problems [2]. The developing brain is particularly vulnerable to hypoxic injury (HI), which results in significant damage in different areas of the central nervous system (CNS) such as the hippocampus, neostriatum, substantia nigra, and cerebellum [3]. The cerebellum is particularly vulnerable to hypoxia due to its long developmental trajectory, which extends from the early embryonic period to the first postnatal weeks in rats and years in humans [4]. Purkinje cell death and reduction in cerebellar volume have been reported after fetal hypoxia/asphyxia occurrence in humans [5], sheep [6], rabbit [7], murine [8], and chicken [9] models. The main disruptive mechanisms implicated in brain injury after hypoxia include energy failure, excitotoxicity, free radical damage, inflammation, and cell death [10].

Neuroprotective actions of growth hormone (GH) have been reported in several models of brain trauma, stroke, spinal cord injury, impaired cognitive function, excitotoxicity, and perinatal hypoxia [11,12]. The molecular and cellular mechanisms through which GH protects the brain involve neurogenesis, neural migration and maturation, synaptogenesis, and anti-apoptotic effects [13]. In the embryonic chicken cerebellum, both GH and GH-receptor (GHR) genes are locally expressed, mainly in the Purkinje and granular cell layers, and an increase in their levels has been reported under hypoxic conditions [14]. Additionally, GH gene knock-down with small interfering RNAs (siRNAs) demonstrated that GH can act as an endogenous neuroprotective factor in response to HI and that this effect was partially mediated through IGF-1 expression, since there was a substantial increase in cell death in chicken cerebellar cell cultures [15] after GH knock-down treatment. In this model, it was shown that GH exerted its antiapoptotic actions through the activation of the PI3K/Akt signaling pathway and an increase in the antiapoptotic protein Bcl-2 [16]. To date, information on the protective and regenerative effects of exogenous GH treatment on cerebellar maturation after hypoxia is limited. Furthermore, it is still controversial whether GH can cross the blood–brain barrier (BBB) and reach the cerebellum, among other brain areas. The presence of GH in cerebrospinal fluid and brain homogenates has been reported after intravenous administration of radioactive GH labeled with ^125^I [17]. In this context, different mechanisms by which GH may reach the CNS have been proposed, including the generation of bioactive fragments [18], passage through the circumventricular organs [19], and translocation by internalization of its receptor in the vascular endothelium of the choroid plexus [20]. Therefore, the aim of the present study was to evaluate the distribution of intravenously added GH in the CNS and the neuroprotective actions of GH in the cerebellum injured by exposure to hypoxia and reoxygenation.

Thus, in this study, we investigated if Cy3-labeled GH can cross the BBB and reach several areas in the brain, both under normoxic and hypoxic injury conditions. It was found that the fluorescent signal was distributed in several brain regions, although more intensely under the hypoxic condition. A similar distribution pattern between labeled GH and GHR in the same areas support the existence of a GH-GHR translocation mechanism, and the co-localization of Cy3-GH and GHR in the cerebellum strata indicates that GH exerts its effects directly. Additionally, we analyzed some of the processes implicated in the neuroprotective role of GH in the embryonic chicken cerebellum exposed to HI. It was found that GH treatment diminished brain damage in hypoxia-injured embryos through mechanisms that involve the inhibition of apoptosis, a reduction in oxidative stress, and the regulation of neurotrophic and inflammatory mediators. Moreover, our results showed that GH protected the injured cerebellum strata by increasing the Purkinje and granular cell survival. This work provides further evidence about the potential use of GH as a neuroprotective and regenerative treatment in perinatal asphyxia.

## 2. Results

### 2.1. Distribution of Cy3-GH in the Chicken Brain Exposed to Hypoxia

Hypoxia was generated in ED15 chicken embryos by wrapping half of the egg’s air chamber with a polyvinyl layer for 24 h to reduce oxygen supply. The hypoxic state was corroborated by analyzing the hypoxia-inducible factor 1-alpha (HIF-1α) immunoreactivity (IR), which increased by 244.3 ± 39.4% after the hypoxic event, in comparison with the normoxic control (100.0 ± 13.0%) (Appendix A).

To assess whether GH crosses the BBB when administered systemically, Cy3-labeled GH (red) was injected into the chorioallantoic vein after 24 h of incubation in normal or hypoxic conditions (Figure 1). Two hours after injection, the tissues were collected and the distribution of Cy3-GH was analyzed in several brain areas.

The results showed that the fluorescent signal was broadly observed in various regions, particularly in the blood vessels (Figure 2, arrowheads), choroid plexuses (Figure 2A,B), periventricular areas (Figure 2C,D), hypothalamus (Figure 2E,F), hippocampus (Figure 2G,H), cortex (Figure 2I,J), and cerebellum (Figure 2K,L), under both normoxic (Nx+GH) and hypoxic (H-Ox+GH) conditions, although the intensity of the signal was apparently higher in the HI condition. To verify that the Cy3 signal (red) observed in Figure 2 corresponded to the labeled GH and not to free Cy3, tissue extracts were analyzed by SDS-PAGE/Western Blot (WB) under reducing conditions, looking for Cy3 and GH-immunoreactivity (GH-IR; Appendix A), and by co-localizing Cy3-GH with GHR by immunohistochemistry (Figure 3). As shown in Appendix A, the WB for Cy3 showed a band of 26 kDa that corresponded to the molecular weight of GH, whereas in the GH immunoblot, a 26 kDa GH-IR band was also found, indicating the integrity of the Cy3-GH in the brain extracts. Liver homogenates were also used as controls (Appendix A). Likewise, although a slight densitometric increase in Cy3-GH fluorescence was observed in liver homogenates under hypoxic conditions (158.7 ± 28.6%), it was not significantly different from the normoxic control (100.0 ± 6.6%) (Appendix A). Although Figure 2 showed that the fluorescent GH signal appeared to be more intense in the choroid plexus, periventricular areas, and cortex under hypoxic conditions (Figure 2B,D,J) in comparison to their respective controls (Figure 2A,C,I), the densitometric results obtained from Cy3-GH WB on the whole brain homogenates showed a slight increase in the hypoxic group that was not statistically significant (174.9 ± 63.8%) in comparison with the control (100.0 ± 11.90%) (Appendix A).

In turn, GH receptor immunoreactivity (GHR-IR) was observed in the same brain areas and with a similar distribution as the Cy3-GH signal. Figure 3 shows the presence of GHR-IR under normoxic conditions (Nx+GH) in the choroid plexus (Figure 3A–C) and in the hypothalamus (Figure 3D) and under hypoxic conditions (H-Ox+GH) in areas such as the hippocampus (Figure 3E) and hypothalamus (Figure 3F). In all these regions, an ample co-localization of GHR-IR with the Cy3-GH signal was observed in many cells (yellow staining and white arrows). In the liver, the immunohistochemical analysis showed that Cy3-GH was located mainly in the hepatocyte periphery (Appendix A, white arrows) under normoxia conditions and in the blood vessels, and within hepatocytes (Appendix A) under hypoxia conditions. Additionally, GHR-IR was widely distributed in the hepatic parenchyma, and a high co-localization with the Cy3-GH signal was observed, under both the normoxic (Nx+GH) (Appendix A) and hypoxic (H-Ox+GH) (Appendix A) conditions.

### 2.2. Distribution of Cy3-GH in the Hypoxic Cerebellum Strata

Before evaluating the protective actions of GH against a hypoxic injury in the embryonic cerebellum, we first determined the distribution of Cy3-GH in the cerebellar strata. Figure 4 shows that 2 h after the microinjection in the chorioallantoic vein, the fluorescent signal was widely distributed in the cerebellum, mainly in the blood vessels (Figure 4B,C, arrowheads), in the cytoplasm of Purkinje cells (Figure 4D,E, arrows), in granular neurons (Figure 4F,G, arrows), and in deep nuclei cells (Figure 4H,I, arrows), both under normoxic (Nx+GH; Figure 4B,D,F,H) and hypoxic (H-Ox+GH; Figure 4C,E,G,I) conditions, although the intensity of the signal was higher in the HI-treated tissues. Furthermore, this increase in the Cy3-GH signal was verified by WB analysis, finding that the fluorescent 26 kDa corresponding to immunoreactive GH increased significantly in the hypoxic cerebellum (281.5 ± 27.36%, H-Ox+GH) in comparison with the normoxic control (100.0 ± 17.35%) (Figure 4J,K) conditions.

On the other hand, as observed by immunocytochemistry, there was a close co-localization between GHR-IR and the Cy3-GH signal in the cerebellar regions, mainly in the deep nuclei, the granular layer (Figure 5A–C, short arrows), and the Purkinje cells, both under normoxia (Nx+GH; Figure 5D, large arrows) and hypoxia (H-Ox+GH; Figure 5E,F, large arrows). These observations show that, as in the brain, systemically administered Cy3-GH reaches various cerebellar layers, keeps its molecular integrity, and interacts with its receptor.

### 2.3. Effects of Hypoxia and GH Treatment on Cerebellar Cytostructure

As depicted in Appendix A, the protective effects of GH against HI were initially analyzed histologycally in the cerebellum of 15ED chick embryos, which after injury, were treated with three daily injections of GH (0.15 µg/g), and hematoxylin-eosin (H-E) staining revealed important morphological changes in the tissue. It was found that, in comparison with the controls, hypoxia provoked evident cyto-structural modifications, as shown by a clear disruption of cellular organization in various cerebellar layers. On the other hand, GH induced significant structural restoration of the layers (Appendix A). Morphometric analysis was used to determine changes in layer thickness and the proportion of dead and alive Purkinje and granule neurons in the H-E stained sections. It was observed that hypoxic (H-Ox+Veh) damage caused a significant decrease in the thickness of cerebellar layers, including the outer granular layer (EGL) (4.8 ± 0.13 µm) and the molecular layer (ML) thickness (53.1 ± 1.14 µm) (Appendix A), as compared with their respective controls (5.4 ± 0.13 µm and 57.2 ± 1.36 µm, Nx+Veh) (Appendix A). In contrast, the thickness of the Purkinje cell layer (PCL) was significantly increased by HI (14.7 ± 0.32 µm, H-Ox+Veh), in comparison with its control (13.2 ± 0.30 µm, Nx+Veh) (Appendix A). Interestingly, systemically administered GH (H-Ox+GH) after HI promoted a significant increase in the thickness of EGL (5.2 ± 0.10 µm) and ML (58.1 ± 1.17 µm), which then showed no differences with the normoxic controls (Appendix A). No statistically significant changes were observed in the PCL thickness in comparison with either the normoxic or hypoxic groups.

The cell number quantification in the cerebellar layers showed that, in comparison with the normoxic control (98.0 ± 0.89%, Nx+Veh), a significant decrease in the survival rate of Purkinje cells (91.1 ± 2.6%) was observed in the hypoxic group (H-Ox+Veh, Appendix A), but GH treatment increased it to 99.2 ± 0.5 % (H-Ox+GH, Appendix A). Likewise, the number of granule neurons decreased significantly (85.5 ± 0.7%, H-Ox+Veh) in the hypoxia group in the ML, as compared with the control (89.01± 0.4% Nx+Veh, Appendix A), while in the GH treated group, the number of granule cells (89.1 + 0.5%, H-Ox+GH) remained similar to that in the control group.

Interestingly, we also found a significant increase (2.2 ± 0.1 cells, H-Ox) in the ectopic localization of Purkinje neurons under hypoxic conditions in the granular layer (GL), compared with the control (1.6 ± 0.1 cells, Nx), while GH treatment after HI showed no difference in ectopic cell number (1.7 ± 0.1 cells, H-Ox+GH) with the control (Appendix A).

### 2.4. Neuroprotective Actions of GH in the Hypoxic Cerebellum

It is known that hypoxia increases apoptosis in neural tissues. We evaluated the antiapoptotic effect of GH treatment after a hypoxic injury in the embryonic cerebellum. As shown in Figure 6, the number of TUNEL-positive cells in the hypoxia-injured tissue (H-Ox+Veh; Figure 6B) increased significantly (294.7 ± 41.7%, Figure 6D) in comparison with the normal control (Nx+Veh; Figure 6A; 100 ± 21.44%; Figure 6D). However, the GH treatment significantly decreased the number of apoptotic cells (H-Ox+GH; Figure 6C) and was not statistically different from the normoxic control (181.4 ± 26.4%; Figure 6D). Likewise, when the Caspase-3 activity was assessed (Figure 6E), we found that, in comparison with the normal control (100 ± 5.2%), hypoxia provoked a significant increase (152.7 ± 8.3%), and GH treatment reversed that deleterious effect (122.1 ± 8.7%).

### 2.5. Effect of GH upon Lipid Peroxidation and iNOS Gene Expression

The hypoxic conditions used in our model induced an expected significant increase in lipid peroxidation (3.1 ± 0.47 nmoles TBARS/mg protein) in comparison with the normoxic control (0.7 ± 0.08 nmoles TBARS/mg protein). Similarly, Figure 6F shows a significant increase in the lipid peroxidation under hypoxic and reoxygenation conditions (2.1 ± 0.22 nM TBARS/mg protein, H-Ox+Veh) compared with the control (0.67 ± 0.13 nM TBARS/mg protein, Nx+Veh), whereas GH treatment significantly reduced the lipid oxidation (1.19 ± 0.21 nM TBARS/mg protein, H-Ox+GH).

Inducible nitric oxide synthase (iNOS) and mitochondrial electron leakage are believed to be the major contributors to ROS/RNS in the immature brain. Thus, we also determined the iNOS mRNA levels by real-time PCR. GH treatment significantly decreased the mRNA expression of iNOS (0.78 ± 0.18-fold, H-Ox+GH) in comparison with the hypoxic group (1.36 ± 0.9-fold, H-Ox+Veh) (Figure 6G).

### 2.6. GH Suppressed Hypoxia-Induced mRNA Expression Levels of Pro-Inflammatory Mediators in the Cerebellum

To investigate the effect of GH treatment on hypoxia-induced inflammation, we used qPCR to measure the expression of mRNAs encoding for the pro-inflammatory factors TNF𝑎, IL-6, and IL-1β. The results showed that hypoxia (H-Ox+Veh) stimulated TNF𝑎 (1.55 ± 0.20-fold H-Ox+Veh, Figure 7A), IL-6 (1.59 ± 0.28-fold, Figure 7B), and IL-1β (1.5 ± 0.28-fold, Figure 7C) mRNA expression, as compared with the corresponding normoxic controls (1.06 ± 0.04-fold, 1.03 ± 0.10-fold, and 0.94 ± 0.09-fold, respectively, Figure 7A–C). In contrast, GH treatment (H-Ox+GH) significantly decreased the mRNA expression of the pro-inflammatory mediators, as follows: TNF𝑎 (0.64 ± 0.10-fold, Figure 7A), IL-6 (0.42 ± 0.08-fold, Figure 7B), and IL-1β (0.38 ± 0.04-fold, Figure 7C).

### 2.7. Differential Modulation of Neurotrophic Factors in Response to GH Treatment in Hypoxic Cerebellum Injury

Figure 8 shows that GH treatment significantly increased the expression of mRNAs encoding for IGF-1 (1.49 ± 0.16-fold; Figure 8A), IGF-1R (1.30 ± 0.08-fold; Figure 8B), VEGF (1.21 ± 0.11-fold; Figure 8C), BDNF (2.0 ± 0.23-fold; Figure 8D), and HIF-1𝑎 (2.52 ± 0.36-fold; Figure 8F) under hypoxic conditions in comparison with the normoxic controls (1.01 ± 0.78-fold, 0.98 ± 0.5-fold, 1.05 ± 0.23-fold, 1.02 ± 0.06-fold, and 1.027 ± 0.12-fold, respectively). In contrast, NT-3 mRNA expression was reduced (0.70 ± 0.18-fold, H-Ox+GH) with GH treatment in comparison with the hypoxia group (1.53 ± 0.09-fold, H-Ox+Veh) (Figure 8E).

### 2.8. Endogenous GH and GHR Expression in Hypoxia Cerebellum Injury

Figure 9A shows that a significant increase (1.81 ± 0.24-fold; H) in GH mRNA expression occurs after 24 h of hypoxia, compared with the normoxic control (1.02 ± 0.09-fold; N). However, as shown in Figure 9B, this effect disappeared when the embryos were exposed to reoxygenation for another 24 h, and no significant changes in GH expression were observed (0.93 ± 0.09-fold) as compared with the control (0.95 ± 0.12-fold). Interestingly, the local expression of GH mRNA was significantly reduced after GH treatment under hypoxic conditions (0.39 ± 0.12-fold, Figure 9B). On the other hand, it was found that GHR mRNA expression was significantly increased both during hypoxia (2.18 ± 0.38-fold; Figure 9C), and after the reoxygenation phase (1.66 ± 0.14-fold; Figure 9D), in comparison with the normoxia control (1.00 ± 0.04-fold and 1.04 ± 0.11; respectively, Figure 9C,D). Furthermore, GH treatment significantly increased (2.07 ± 0.11-fold; Figure 9D) GHR mRNA expression compared with its control.

### 2.9. Endogenous GH and GHR Expression in Hypoxia Cerebellum Injury

Figure 9A shows that a significant increase (1.81 ± 0.24-fold; H) in GH mRNA expression occurs after 24 h of hypoxia, compared with the normoxic control (1.02 ± 0.09-fold; N). However, as shown in Figure 9B, this effect disappeared when the embryos were exposed to reoxygenation for another 24 h, and no significant changes in GH expression were observed (0.93 ± 0.09-fold) as compared with the control (0.95 ± 0.12-fold). Interestingly, the local expression of GH mRNA was significantly reduced after GH treatment under hypoxic conditions (0.39 ± 0.12-fold, Figure 9B). On the other hand, it was found that GHR mRNA expression was significantly increased both during hypoxia (2.18 ± 0.38-fold; Figure 9C) and after the reoxygenation phase (1.66 ± 0.14-fold; Figure 9D) in comparison with the normoxia control (1.00 ± 0.04-fold and 1.04 ± 0.11; respectively, Figure 9C,D). Furthermore, GH treatment significantly increased (2.07 ± 0.11-fold; Figure 9D) GHR mRNA expression compared with its control.

## 3. Discussion

Here, we show that GH passes through the BBB and reaches several CNS areas, including the cerebellum, choroid plexuses, periventricular areas, hypothalamus, hippocampus, and cortex, when administered intravenously, either during normal aeration or under hypoxia conditions in the chicken embryo. In addition, our findings support that GH is capable of exerting protective actions in the hypoxic pathophysiological response, which involves changes in the molecular markers for apoptosis, oxidative stress, inflammation, and the expression of neurotrophic factors in the cerebellum. Taken together, these results provide further information about the mechanisms that mediate the neuroprotective and regenerative effects of GH when the developing cerebellum is exposed to a hypoxic injury.

Whether GH crosses the BBB and exerts effects on the CNS has been controversial. Some studies in rats showed that 10–30 min after peripheral administration, ^125^I-GH was found either in whole brain homogenates and/or in the telencephalon, diencephalon, pons, and cerebellum [17,21]. To date, the cellular mechanisms by which GH passes through the BBB are far from being fully elucidated; however, similarities in the molecular weight of GH with other hormones such as prolactin suggest that GH could reach the CNS after binding to its receptor in the choroid plexuses or through a saturable transport independent of its receptor [22,23]. In contrast, Pan et al., (2005) [21] did not find a saturable transport system when administering 1 µg of unlabeled mouse GH and proposed that GH crossed BBB through passive diffusion rather than by active transport. Furthermore, Brown et al., (2013) [23] reported that 1 mg of mouse prolactin was necessary to block its transport, a reason why studies using different GH concentrations are important. In contrast, Horvat et al., (1982) [24], after the administration of ^125^I-GH i.v. (30 min), suggested that GH probably did not cross the BBB, given the low radioactivity levels found in the tonsil and thalamus in comparison with those detected in the kidney, liver, and pituitary. Additionally, it was shown that most of the i.v. administered GH was located in peripheral tissues such as the liver, diaphragm, and kidney, compared with that found in the CNS [17]. Our results in the ED15 chicken embryo give evidence that intravenously injected GH crossed the BBB and reached several areas of the CNS since at that stage of development it is already functional [25,26]. Although some reports describe that the BBB is not yet fully formed in 15-day chicken embryos, studies indicate that junctional complexes between endothelial cells are present from very early stages [26] and have shown that circulating horseradish peroxidase can be excluded from the brain by 10 days in ovo, indicating that the chicken BBB can restrict the passage of macromolecules between 10 and 16 days in ovo and matures completely two days after hatching [25]. Fleming et al., (2016) [27] demonstrated that the Cy3-labeled GH injected into the chorioallantoic vein of 15-day embryos was internalized in the retinal ganglion cells, suggesting that GH passes through the blood–retinal barrier.

Furthermore, here, we show that GH was capable of reaching the CNS under either the normoxia or hypoxia conditions. After 2 h of administration, Cy3-GH was observed in the brain, mainly in the blood vessels, choroid plexuses, periventricular areas, and cerebellum and with less intensity in the hypothalamus, hippocampus, and pallium. Additionally, Cy3-GH was found in sections of the liver, which was used as control tissue due to its high blood supply and the high density of GHRs.

Our findings showing that labeled GH was significantly increased in the brain and cerebellum when the animals were exposed to hypoxia, in comparison with the control group, suggest that the BBB could be damaged or that its permeability is modified under this condition. In murine models of hypoxia-ischemia, an increase in permeability was observed for both high and low molecular weight molecules, and this was associated with alterations in the expression of tight junction proteins and endothelial cells in the BBB [28,29]. Mustafa et al., (1995) [30] observed that the permeability to GH increased in the spinal cord, cortex, and cerebellum, 2 h after inducing damage at the T9-T12 level, but it was inhibited by the administration of an antioxidant (H 290/51), which reduces lipid peroxidation. On the other hand, Scheepens et al., (1999) [31] observed an increase in the immunoreactivity of GHR and/or GHBP (GH binding protein) in the blood vessels of the frontoparietal and somatosensory cortex as well as in the choroid plexuses one hour after hypoxic-ischemic damage in the rat. Among the mechanisms proposed to participate in GH transport through the BBB is the translocation of its receptor in the vascular endothelium through the choroid plexus [20] or active transport [21]. It has been suggested that the increase in GHR could facilitate GH transport through the BBB [31]. The co-localization of a similar distribution pattern between the GH receptor (GHR) and GH could support this hypothesis. The presence of GHR has been reported in the cerebellum, hippocampus, hypothalamus, choroid plexus, periventricular areas, and the ventricular line in the rat brain [32], which is consistent with our findings of the co-localization of GH and its receptor in several chicken embryonic brain areas. Thus, the results of this work add further information to previous studies regarding the notion of the existence of a GH-GHR translocation process to reach the CNS and suggest that it could be involved in a potential treatment for the hypoxic-ischemic injury.

In the cerebellum, the distribution of Cy3-GH was observed mainly in the blood vessels, Purkinje cells, granular layer, and deep nuclei cells. These results agree with those reported previously, where it was found that the highest immunoreactivity for both GH and its receptor occurred in the Purkinje cells and, to a lesser extent, in granular cells and neural processes in the molecular layer of the 4-week-old chicken cerebellum [14]. Additionally, Lobie et al., (1993) [32], reported a moderate GHR immunoreactivity in the outer and inner parts of the granular layer of the cerebellum. Our findings, together with previous reports, suggest that the cerebellum is a target structure of GH actions [14,16,32].

Previous studies have demonstrated that hypoxic-ischemic insults interfere with the normal development of the cerebellum [4,9]. Lee et al., (2001) [9], demonstrated that, in chick embryos, chronic hypoxia (from ED2 to ED21) hinders the normal development of Purkinje cells marked by small cell size, poorly developed dendrites, low cell density, and ectopia. Similar findings were reported in the developing mammalian cerebellum [4,33,34]. Biran et al., (2011) [4], described a decrease in the number of Purkinje cells, granular neurons, and interneurons as well as in the thickness of molecular (ML) and granular (GL) layers in two-day-old rat pups exposed to global hypoxia or hypoxic-ischemia injury. In this work, we found that hypoxia disrupted the cerebellar cytoarchitecure and caused a decrease in the thickness of the external granular layer and molecular layer. Additionally, as shown previously [9], an increase in the PCL thickness, associated with greater dispersion and ectopic distribution of the Purkinje cells, was observed. The high susceptibility of Purkinje cells to hypoxia may be associated with their high metabolic demand and elaborated synaptic interactions [34]. The ectopic distribution of Purkinje cells has been reported in some mutant mice with cerebellar abnormalities, e.g., reel mice [33] and normal rodents [35]. Thus, our results support the notion that hypoxia disturbs the migration of Purkinje cells on their way to the cerebellar cortex. The GH-induced recovery of cerebellar layer thickness observed in this study was mainly located in three regions: EGL, ML, and PCL, which were associated with a significant decrease in granular and Purkinje cell death induced by HI. These results are consistent with previous reports demonstrating the antiapoptotic and restorative effects of GH against hypoxic-ischemic injury in primary cultures of chicken granular neurons [15,16] or after excitotoxic damage in neuroretinal cells [36]. To our knowledge, this is the first report about GH-induced protection of Purkinje cells.

The cerebellum is a highly conserved structure among vertebrates and is well known for its important role in motor coordination [37]. GH is involved in the growth of the cerebellum since GH-deficient mice presented a 20% reduction in the size of this organ in comparison with normal mice [38]. The early administration of GH during the first 20 days after birth restores the reduction in the size of the brain in GH-deficient mice by 18% [39]. In the chicken, the local expression of GH in the Purkinje cell bodies and the primary dendritic trunks after hatching coincides with the growth of their dendritic trees and with the number of synapses formed after the last stage of embryogenesis [14]. In cerebellar primary cell cultures, the local expression and/or the exogenous administration of GH and IGF-1 were shown to be important in maintaining cell survival during an acute hypoxic-ischemic injury [15,16]. In this work, a significant increase in apoptosis was found after hypoxia and 24 h of reoxygenation. This is consistent with previous studies showing that an important rise in TUNEL signal was found in the brain of animals exposed to hypoxia [16] using the same protocol and in studies carried out in models of unilateral hypoxia-ischemia in postnatal rats where the highest peaks of caspase-3 activity and TUNEL signal were observed 24 h after damage [40,41,42]. The main TUNEL signal occurred in the inner granular layer and to a much lesser grade in the outer granular layer, which coincides with several reports showing that the majority of death by apoptosis is observed in these layers, while the Purkinje cells appear to die primarily from necrosis [43]. Regarding caspase-3 activity, our results agree with previous works [15,16,44], in which a significant increase in this enzyme was observed after HI damage; however, when GH treatment (150 µg/kg i.v.) was administered after injury, a clear decrease in apoptosis occurred [15,16]. As for the results obtained with the TUNEL assay, they are in accordance with those previously obtained in the laboratory in primary and organotypic chicken neuronal cultures [16] and in in vivo studies in hypoxia-ischemia models in rats [44,45]. Likewise, GH treatment significantly decreased the number of TUNEL-positive cells in chick embryo retinal explant cultures by lowering the mRNA expression of caspase-3 and the apoptosis-inducing factor (AIF) [46]. In these studies, an increase in Bcl-2 and a decrease in Bax proteins were observed, resulting in a decrease in the Bax/Bcl-2 ratio, which may result in the inhibition of caspase-3 [15,16]. Other studies have also shown that the administration of several doses of GH exerts antiapoptotic, neuroprotective, and regenerative actions in response to injury [45,46,47,48,49]. Interestingly, the administration of a specific anti-cGH antibody was capable of blocking the neuroprotective actions of GH (as determined by measuring its effects upon cell survival, caspase-3 activity, and neurite outgrowth) in embryonic chicken pallial cultures exposed to HI injury [50].

Here, we also evaluated the effect of GH on oxidative stress and inflammation markers [10]. As previously mentioned, the immature brain is highly susceptible to hypoxic damage, especially the periventricular white matter [51], where the direct involvement of microglial cells in the pathophysiological response has been described. Microglial cell activation is associated with a cascade of events that includes increased glutamate release, inducible nitric oxide synthetase (iNOS) expression, and nitric oxide (NO) production, as well as the release of the pro-inflammatory cytokines TNF-𝑎 and IL-1β, and a diminished release of neurotrophic factors such as IGF-1 and IGF-2 [52]. In the cerebellum, hypoxic damage is primarily associated with Purkinje cell death and a decrease in the thickness of the molecular and granular layers during development. Although the precise mechanisms responsible for Purkinje cell death as a result of hypoxic damage are not clearly understood, recent work shows that microglial cells induce Purkinje cell death by increasing the release of TNF-α and IL- 1β in the global hypoxic rat model (exposure to 8% O_2_ for 2 h) [53]. In the same experimental paradigm, Yao et al., (2013) [36]. identified that Toll-like receptor 4 (TLR4) is a mediator in the activation of microglia and the production of inflammatory factors in the cerebellum after hypoxia. The observed decrease in the mRNA expression of the main pro-inflammatory factors TNF-𝑎, IL-6, IL-1β, and iNOS induced by GH addition indicates an anti-inflammatory effect of this hormone in the injured cerebellum. Additionally, GH treatment has been shown to decrease the production of TNF-𝑎 and IL-1β in murine and human cell lines of macrophages/monocytes [54,55,56].

The neonatal brain, with its high concentrations of unsaturated fatty acids and low concentrations of antioxidants, is particularly vulnerable to reactive oxygen and nitrogen species (ROS and RNS) damage. These powerful oxidizing and nitrating agents can cause direct damage to proteins, DNA, and lipids. After hypoxia, the concentrations of malondialdehyde (MDA, a lipid peroxidation indicator) increased in the plasma of human perinatal fetuses [57] and newborn infants [58]. Hypoxia also induced an increase in MDA in the liver, brain, and heart of chicken embryos [59]. It has been reported that hypoxia stimulates the inflammatory response by upregulating the early response gene, inducible nitric oxide synthase (iNOS), which in turn leads to the rapid overproduction of NO. Excess NO has been shown to cause lipid peroxidation and, indirectly, other oxidative damage when combined with superoxide to form peroxynitrite (ONOO^−^), which is then cleaved to form free radicals such as OH^−^ and NO_2_^+^ [60]. The observed reduction in lipid peroxidation and the iNOS mRNA expression suggests a plausible mechanism of action of GH treatment upon oxidative stress damage. Our findings support previous reports showing that GH administration provoked a decrease in inducible and endothelial iNOS immunoreactivity in the HI-injured rat brain, thus indicating that GH exerts a neuroprotective role in response to oxidative stress and inflammation [45].

It has been proposed that GH and IGF-1 may play crucial roles in the autocrine/paracrine regulation of proliferation, differentiation, survival, and migration of both neural progenitors and mature neurons during the development of the CNS. Previously, our group reported that the local expression of GH and IGF-1 mRNAs was significantly increased in the HI-injured cerebellar tissue, and this resulted in a neuroprotective response against damage [14,15]. Our findings here, regarding the increase in GH and GHR mRNA expression in the cerebellum of the chicken embryo exposed to hypoxia, support the hypothesis that the neuroprotective response observed involves an important role of GH through autocrine/paracrine mechanisms, in addition to its endocrine contribution.

Several reports indicate that the protective action exerted by GH in response to diverse types of neural damage may implicate the participation of other neurotrophic peptides, such as IGF-1, BDNF, NT-3, VEGF, EPO, and the transcription factor HIF-1α [14,15,36,49,61,62,63,64]. For instance, GH administration attenuated cognitive deficits in juvenile rats exposed to chronic and intermittent hypoxia, and this was associated with activation of IGF-1, EPO, and VEGF-A mRNAs in the hippocampus [49]. Additionally, GH was shown to induce BDNF and NT-3 expression while having protective actions in the chicken neural retina exposed to kainate excitotoxic damage [61]. In this work, we found that GH stimulated the expression of IGF-1, IGF-1R, VEGF, BDNF, and HIF-1α mRNAs in the cerebellum of chicken embryos after hypoxia-reoxygenation injury. The increase in IGF-1 mRNA was analogous to that found previously after the addition of GH in HI-injured cerebellar cell cultures [15]. These results agree with observations in cortical neuron cultures obtained from P7 mice subjected to neonatal acute hypoxia, where rhGH increased EPO, IGF-1, IGF-2, IGF-BP, and VEGF after 48 h of regeneration period, whereas in the developing brain, the GH treatment increased the levels of the EPO protein, exerted anti-inflammatory effects by modulating IL- 1β and TNF-α, and promoted BBB stabilization [65]. On the other hand, the results showed that NT-3 mRNA expression was diminished after GH treatment. It has been postulated that NT-3 does not promote survival in cerebellar granule neurons possibly due to low levels in TrkC expression [66], suggesting a different neurotrophic action during hypoxia. Taken together, the observed effects of GH upon the local neurotrophic factor system in response to hypoxia suggest the existence of a complex mechanism mediating the neuroprotective actions attributed to this hormone, which deserves further research.

In summary, our findings confirm that GH can cross the BBB and reach several areas of the CNS after a hypoxic injury. The similar distribution pattern between labeled GH and its receptor in various brain areas supports the existence of a GH-GHR translocation mechanism, and the co-localization of Cy3-GH and GHR in the cerebellum strata suggests that GH exerts its effects directly. However, further studies are required to elucidate the precise mechanisms by which GH is transported through the BBB. Additionally, it was shown that the administration of exogenous GH diminished cerebellar damage in hypoxia-injured embryos through mechanisms that involve the inhibition of apoptosis, a reduction in oxidative stress, and the regulation of neurotrophic and inflammatory mediators.

Currently, hypothermia is the most common therapy used for hypoxic-ischemic encephalopathy (HIE) in neonates; however, it is effective only within the first 6 h and in moderate impact injuries [67]. Thus, it is desirable to develop new treatment alternatives that combine physical and pharmacological approaches. In this regard, it has been shown that GH administration improves cognitive function after intermittent hypoxia or stroke in rats [49,68]. Additionally, in humans, stroke and traumatic brain injury (TBI) decrease the levels of IGF-1, but GH therapy improves the speed of processing and memory and decreases the severity of depression, improving the quality of life of patients [69,70]. The results in children and adults report safe profile effects of long-term therapy [69,71], although more studies are needed to analyze the effectivity of GH treatment and IGF-1 levels in perinatal asphyxia, and its safety profile in newborns. Overall, the results from this work provide further evidence about the potential of GH to be considered as a neuroprotective and neuroregenerative complementary therapy for HI encephalopathy.

## 4. Materials and Methods

### 4.1. Animals

Pathogen-free, fertilized chicken eggs (Gallus gallus domesticus, White Leghorn breed) at 15 days of embryogenesis (15 DE) were obtained from Pilgrim’s Pride (Querétaro, Mexico) and incubated at 39 °C in a humidified air chamber (IAMEX, Querétaro, Mexico). The eggs were rotated one-quarter of a revolution every 50 min during incubation. After treatments, the chicken embryos were sacrificed by decapitation according to a protocol approved by the bioethical committee of the Instituto de Neurobiologia, UNAM, and under the Mexican official regulation (NOM-062-ZOO-1999).

### 4.2. Treatments

Initially, the chorioallantoic vein of the embryos was exposed by cutting a window (5 × 5 mm) in the eggshell with a drill in a dark room, using a lamp. Then, experimental hypoxic conditions were induced by enveloping half of the chicken embryo air chamber with a polyvinyl layer for 24 h at 39 °C, to limit gas exchange. Hypoxic conditions were terminated by removing the polyvinyl layer, and then, the embryos were further re-oxygenated for 24 h at 37 °C (H+Ox). The normoxic (Nx) control group was incubated under normal air conditions at 37 °C for 48 h. At the end of the first 24 h period, the embryos were microinjected through the chorioallantoic vein with 100 µL of either Cy3-labeled GH (Cy3-GH; 0.15 µg/g of body weight, for 2 h, to determine BBB crossing and distribution within several CNS areas, Figure 1A) or recombinant chicken GH (rcGH, American Cyanamid AC 4797-100, Wayne, NJ, USA, 0.15 µg/g) to study the neuroprotective effects 24 h after hypoxia/reoxygenation injury by analyzing apoptosis, lipid peroxidation, inflammatory mediators and the expression of several neurotrophic factors (Figure 1B). The control groups received 100 µL of a saline solution as a vehicle.

Cy3 conjugation to GH (Cy3-GH) was performed according to the Cy3 Mono-Reactive dye pack manufacturer’s instructions (GE Healthcare Life Sciences, Amersham, ON, Canada).

### 4.3. Histological Analysis

To evaluate the effect of hypoxia injury and GH treatment upon tissue cytoarchitecture, cerebellums were collected and fixed in Carnoy´s solution for 24 h at 4 °C, dehydrated in ethanol, and embedded in paraffin wax. Tissue sections (10 µm) were cut using a microtome and mounted onto charged slides [14]. For histological analysis, slides were stained with hematoxylin-eosin (H-E) [57]. Cerebellum cell-layer thickness was determined in slices stained with H-E in at least 10 microscopic fields per animal, and 3 individual animals per experimental group were examined. Morphometrical analysis was performed in the same area of the folia for equivalent group comparisons. Images were captured using an Olympus BX51 fluorescence microscope (Tokyo, Japan) and analyzed with Image Pro 10 software (Media Cybernetics, Rockville, MD, USA).

### 4.4. Immunohistochemistry

Embryonic brains or cerebellums were collected and fixed in Zamboni (4% paraformaldehyde, 1.5% picric acid in 0.02 M PBS) solution for 24 h and cryoprotected on 20% sucrose at 4 °C. Tissue sections (30 µm) were cut using a freezing microtome (SM2000R, Leica Microsystem, Wetzlar, Germany) and mounted onto charged slides.

To evaluate the distribution of Cy3-GH within several brain areas, slides were incubated with 500 ng/mL of DAPI (4′,6-diamidino-2-phenylindole, D9542, Sigma Aldrich, Burlington, MA, USA) for 30 min in the dark, to label cell nuclei. After washing for 30 min with TBS, all the slides were mounted with vectashield (Vector Laboratories Inc., Burlingame, CA, USA), and images were captured with a Zeiss LSM 780 DUO (Carl Zeiss AG, Oberkochen, Germany) confocal microscope.

To evaluate specific interaction and distribution of Cy3-GH, slides were incubated with anti-GHR. The slides were washed 3 times with PBS, and then, free binding sites were blocked with 5% non-fat dry milk (Bio-Rad, Hercules, CA, USA) for 2 h. After blocking, the slides were washed with TTBS (0.1% Triton X-100 in TBS) 3 times and then incubated overnight with the mouse anti-GHR antibody (MA1-82444, Thermo Scientific Pierce Antibodies, Waltham, MA, USA, Table 1). Then, the slides were washed (3 × 10 min) and incubated with the secondary antibody rabbit anti-mouse IgG-FITC (Table 1). After 3 washes, the cell nuclei were labeled with DAPI, as mentioned above. Slides were mounted with vectashield and images were captured with a Zeiss LSM 780 DUO confocal microscope.

### 4.5. Western Blot Analysis

To evaluate the structural integrity of Cy3-GH, the slides were incubated with anti-GH and anti-Cy3 antibodies. Brain tissues were homogenized with an ultrasonicator (GE 130PB, Cole-Parmer, Vermont Hills, IL, USA) in 50 mM HCl-Tris, pH 9.0 buffer, containing a protease inhibitor cocktail (Mini-complete, Roche, Basel, Switzerland). The supernatants were collected after centrifugation of the extracts (12,500 rpm, 15 min), and total proteins were determined with the Bradford assay (Bio-Rad, Hercules, CA, USA). Samples (80 µg/lane) were separated in 12.0% SDS-PAGE gels under the reduction condition and then transferred onto nitrocellulose membranes (Bio-Rad, Hercules, CA, USA), as previously described [16]. Nitrocellulose-free binding sites in the membranes were blocked with 5% non-fat dry milk (Bio-Rad, Hercules, CA, USA) in TBS (Gibco, Grand Island, NY, USA) for 2 h, at RT. Membranes were then incubated overnight at 4 °C with the corresponding specific primary antibodies (Table 1), diluted in TTBS (0.05% Tween (*v*/*v*) in 1 × TBS). After washing the membranes with TTBS (3 × 10 min), they were incubated for 2 h, at RT, with the corresponding HRP-conjugated secondary antibodies (Table 1). Immunoreactive bands were developed using an ECL blotting detection reagent (Amersham-Pharmacia, Buckinghamshire, UK) on autoradiography film (Fujifilm, Tokyo, Japan). Kaleidoscope molecular weight markers (Bio-Rad, Cat. 1610375) were used as a reference for molecular mass determination. To study the difference among treatments in the same blots, a stripping method was used [16]. The intensity of HIF-1a, Cy3, and GH immunoreactive bands was normalized compared to the actin band intensity used as the protein loading control. Immunoblotting experiments were repeated 3 times, scanned, and quantified by densitometric analysis using ImageJ software (developed by NIH, freeware [72]).

### 4.6. Determination of Lipid Peroxidation (LPO)

Lipid peroxidation in cerebellar homogenates was determined following the method described by Ohkawa et al., (1979) [73]. Briefly, 500 µg of protein samples was mixed with 0.2 mL of 8.1% SDS, 1.5 mL of 20% glacial acetic acid, and 1.5 mL of 0.8% thiobarbituric acid (TBA). Then, the tubes were mixed and heated to 95 °C for one hour on a water bath and cooled under tap water before mixing with 1 mL of distilled water and 5 mL of n-butanol. The organic phase was separated by centrifugation at 2200× *g* for 10 min, and its absorbance at 532 nm was evaluated in a spectrophotometer (Varioskan flas, Thermo Fisher Scientific, S.L.) to determine the amount of malondialdehyde (MDA) formed, using appropriate controls. The results were expressed as nmol thiobarbituric acid reactive substances (TBARS)/mg protein.

### 4.7. Evaluation of Apoptosis

The enzymatic activity of caspase-3 in the cerebellum was determined using a colorimetric assay kit (Assay Designs Inc., Ann Arbor, MI, USA). Tissues were homogenized in 250 µL of lysis buffer (HEPES/NaOH 10 mM pH 7.4, EDTA 2 mM, DTT 1 mM, CHAPS 0.1%). Samples (12 µg of protein) from each treatment, standards, *p*-nitroaniline (pNA) standard, and blank controls were placed in a 96-well plate. After 3 h of incubation at 37 °C, the reaction was stopped with 1 N HCl and then read at 405 nm in a microplate reader (Bio-Rad, Mod. 550, Hercules, CA, USA). The caspase-3 enzymatic activity was calculated as units per microgram of protein [16], normalized, and expressed as percent activity relative to the 100% normoxia as a control (Nx+Veh) group.

Apoptosis was also measured using the APO-BrdU TUNEL assay kit (11684795910, Roche, Boston, MA, USA). Cerebellums were collected, fixed in Zamboni solution for 24 h, and then cryoprotected on 20% sucrose at 4 °C. Tissue sections (30 µm) were cut using a freezing microtome (SM2000R, Leica Microsystem, Wetzlar, Germany) and mounted onto charged slides. Cerebellum slides were washed in PBS (3 times) and permeated with citrate buffer (10 mM sodium citrate, pH 6.0, and 1% triton) at 80 °C for 30 min. After 2 washes in PBS, the slides were treated with proteinase K (2 µg/mL in 10 mM TRIS/HCL pH 7.4 buffer) for 10 min at 37 °C and washed with PBS. Slides were then incubated in 1% H_2_O_2_ in 50% methanol to block endogenous peroxidase and then washed in PBS. Then, they were incubated with 50 µL of the reaction mixture from the TUNEL assay kit (50 µL of TdT and 450 µL of dUTPs solutions) for 1.5 h at 37 °C in a humid chamber. The slides were washed for 30 min in PBS and analyzed with a fluorescence microscopy (ApoTome, Carl Zeiss, Germany). TUNEL labeling was quantified using Image J Software (National Institutes of Health, EUA). The proportion of apoptosis was determined by expressing the number of TUNEL-positive cells detected in the field as a percentage of the total number of cells counterstained with DAPI (4′-6-diamino-2-phenylindole dihydrochloride), a marker of the cell nuclei. The quantification was made in 10 separate fields for each slide.

### 4.8. RT-PCR

Total RNA was extracted using 200 µL of the TRIZOL reagent (Invitrogen, Waltham, MA, USA) and purified using a Direct-zol TM RNA MiniPrep kit with Zymo-SpinTM, according to the manufacturer’s directions (Zymo Research Corp., Irvine, CA, USA). The total amount of RNA was quantified using a NanoDrop-1000 spectrophotometer (Thermo Scientific, Waltham, MA, USA). The cDNA was synthesized from 1 µg of total RNA using 200 U of M-MLV Reverse Transcriptase (Invitrogen Cat no. 28025-013, Waltham, MA, USA), 25 mM dNTPs, 0.2 µg oligo d(T), and 0.2 µg random hexamers, for 60 min at 37 °C.

### 4.9. Quantitative PCR (qPCR)

The expression of GH, GH receptor (GHR), IGF-1, and IGF-1 receptor (IGF-1R) mRNAs were quantified by quantitative PCR (qPCR) in a Step One Real-Time PCR system (Applied Biosystems, Foster, CA, USA), and using SYBR Green (Roche, Mannheim, Germany) in 10 µL final volume containing 3 µL cDNA (diluted 1:20 for 18S and 1:3 for the other genes, Table 2), and 1 µL of each specific primer (0.5 µM). The primer sets used (Table 2) were designed to amplify avian mRNAs and to cross intron–exon boundaries to control for genomic DNA contamination. Reactions were performed under the following conditions: initial denaturation at 95 °C for 10 min; followed by 40 cycles at 95 °C for 15 s, 60 °C for 30 s, and 72 °C for 30 s. Dissociation curves were included after each qPCR experiment to ensure primer specificity. The relative abundance of the studied mRNAs was calculated using the comparative threshold cycle (Ct) method and employing the formula 2-ΔΔCT [74], where the quantification is expressed relative to the geometric mean of 18S mRNA [75].

### 4.10. Statistical Analysis

In all experiments, values are expressed as mean ± error standard (SEM). Significant differences between multiple groups were determined by one-way ANOVA and Tukey’s post hoc test. Unpaired Student’s *t* test was used for comparison between two groups where appropriate. Groups with different letters are significantly different from each other, *p* values less than 0.05 were determined to be statistically significant (* *p* < 0.05; ** *p* < 0.01; *** *p* < 0.001).

## Figures and Tables

**Figure 1 ijms-23-11546-f001:**
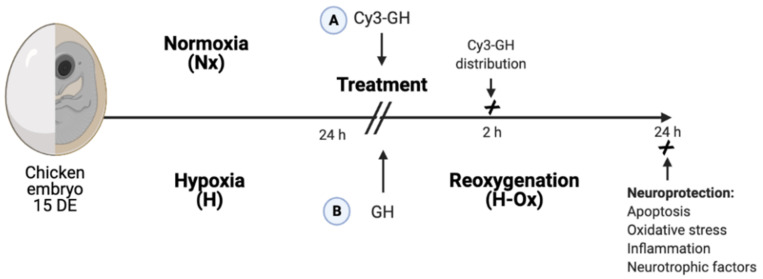
Experimental design. Initially, 15-day old chicken embryos were exposed to either normoxia or hypoxia (by wrapping half of the eggshell with a polyvinyl membrane) for 24 h, at 39 °C. Then, two approaches were followed: (**A**) Cy3-GH (0.15 µg/g) was microinjected in the chorio-allantoid vein, and 2 h later, the distribution of the fluorescent hormone and its integrity were evaluated in several brain regions and in the liver by histochemistry and Western blot; (**B**) embryos were treated with recombinant chicken GH (0.15 µg/g) and were submitted to re-oxygenation for another 24 h, and its effects upon apoptosis (caspase-3 and TUNEL assay), oxidative stress (lipid peroxidation), and expression of inflammatory mediators and neurotrophic factors were evaluated.

**Figure 2 ijms-23-11546-f002:**
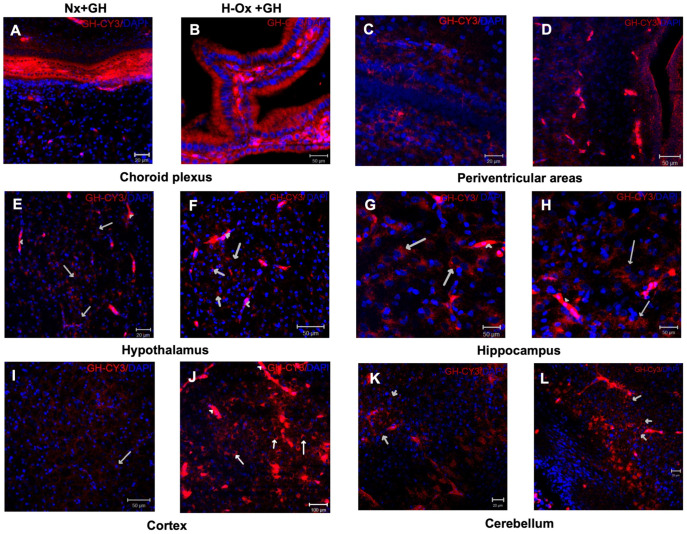
Distribution of Cy3-GH signal in several brain areas in the chicken embryo. Representative images showing the presence of Cy3-GH in the choroid plexus (**A**,**B**), periventricular areas (**C**,**D**), hypothalamus (**E**,**F**), hippocampus (**G**,**H**), cortex (**I**,**J**), and cerebellum (**K**,**L**) 2 h after its administration through the chorio-allantoid vein. The images correspond to animals that were maintained in normoxia (**A**,**C**,**E**,**G**,**I**,**K**) and animals exposed to hypoxia (**B**,**D**,**F**,**H**,**J**,**L**). The arrows indicate the presence of Cy3-GH in cells, while the arrowheads indicate vessels where the fluorescent hormone was found. Scale bar 20 µm in (**C**,**E**,**K**,**L**); 50 µm in (**A**,**B**,**D**,**F**,**G**,**H**,**I**) and 100 µm in (**J**).

**Figure 3 ijms-23-11546-f003:**
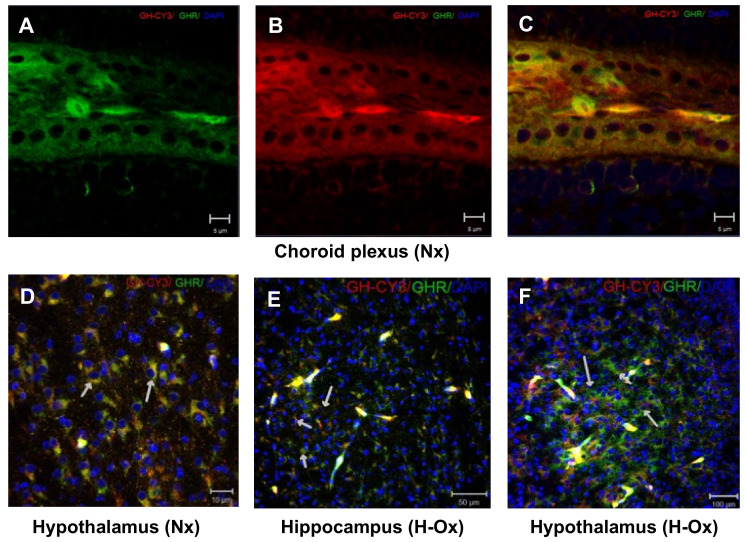
Co-localization of Cy3-GH and GH receptor (GHR)-immunoreactivity in the embryonic chicken brain. Representative immunohistochemical images showing the distribution of (**A**) GHR-IR (green), (**B**) Cy3-GH (red), or (**C**) merge of both (yellow) in the choroid plexus; hypothalamus (**D**,**E**); and hippocampus (**F**) of animals that were treated with Cy3-GH under either normoxia (**A**–**D**) or hypoxia conditions (**E**,**F**). The yellow regions, indicated by arrows, show the co-localization of the hormone with its receptor. Cell nuclei were stained with DAPI (blue). Scale bar 5 µm *(***A**–**C**); 10 µm in (**D**); 50 µm in (**E**), and 100 µm in (**F**).

**Figure 4 ijms-23-11546-f004:**
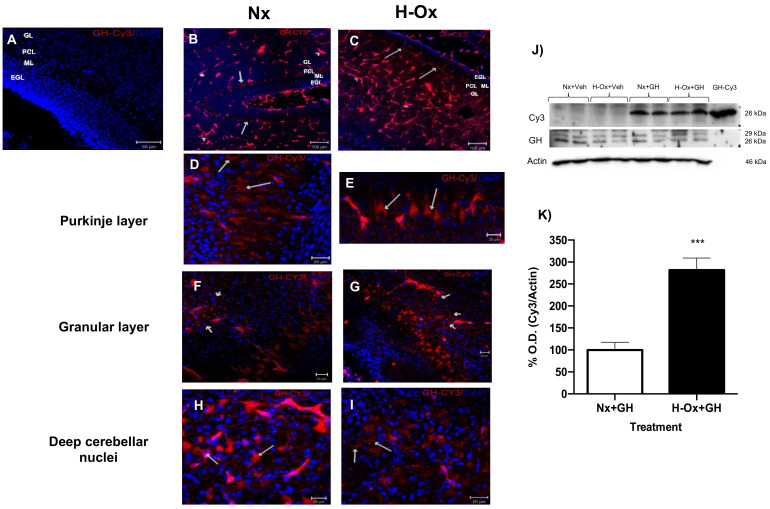
Distribution of Cy3-GH in cerebellar layers after 2 h of administration through the chorio-allantoic vein of chicken embryos. (**A**) Control animals that received only vehicle (saline solution); (**B**,**D**,**F**,**H**) animals injected with Cy3-GH but maintained under normoxia; and (**C**,**E**,**G**,**I**)) animals exposed to hypoxia injected with Cy3-GH. Arrows indicate the presence of Cy3-GH in cerebellar cells in the following regions: (**D**,**E**) Purkinje layer, (**F**,**G**) granular layer, and (**H**,**I**) deep nuclei. The integrity of the Cy3-GH was validated by Western blot in cerebellar homogenates. The added fluorescent hormone (Cy3) had a similar molecular weight (26 kDa) as the GH-IR band under both normoxic and hypoxic conditions. As expected, no Cy3 band was observed in animals that received only vehicles. Nx+Veh (n = 3), H-Ox+Veh (n = 3), Nx+GH (n = 5), H-Ox+GH (n = 4). (**J**,**K**) Densitometric analysis of WB shows the relative proportion of Cy3-GH found in cerebellar extracts under normoxia (Nx) and hypoxia (H-Ox) conditions. Bars represent mean ± SEM. Asterisks (***) indicate significant differences *p* < 0.001, by Student’s *t*-test. Abbreviations: EGL: external granular layer, ML: molecular layer, PCL: Purkinje cell layer, GL: granular layer. Scale bar 20 µm in (**D**–**I**); 50 µm in (**A**), and 100 µm in (**B**,**C**).

**Figure 5 ijms-23-11546-f005:**
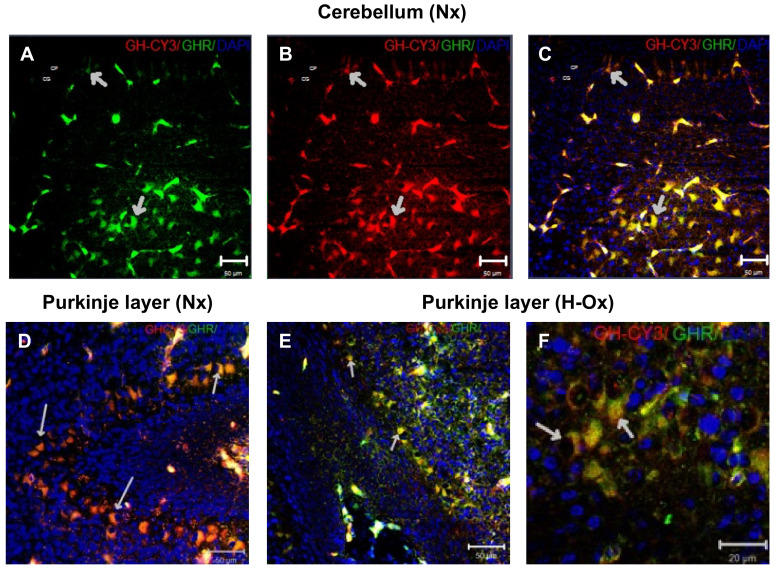
Co-localization of Cy3-GH and GH receptor (GHR)-immunoreactivity in the cerebellum. Representative immunohistochemical images showing the distribution of (**A**) GHR-IR (green), (**B**) Cy3-GH (red), or (**C**) merge of both (yellow) in the cerebellar deep nuclei and in the Purkinje cell layer (**D**–**F**) of embryos that were treated with Cy3-GH under either normoxic (**D**) or hypoxic (**E**,**F**) conditions. The yellow regions, indicated by arrows, show the co-localization of the hormone with its receptor. Cell nuclei were stained with DAPI (blue). Scale bar 20 µm in (**F**); 50 µm in (**A**–**E**).

**Figure 6 ijms-23-11546-f006:**
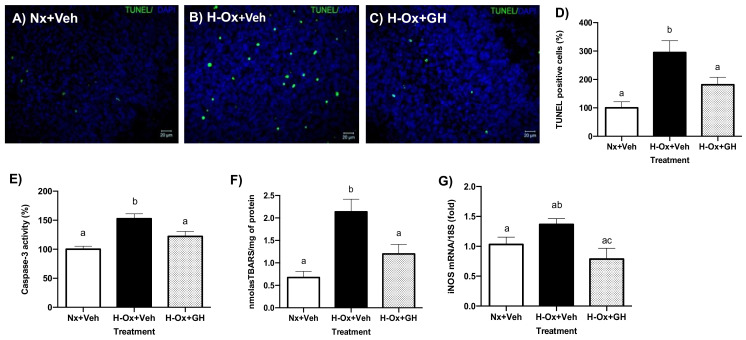
Protective effects of GH treatment on apoptosis and oxidative stress induced by hypoxia injury in embryonic cerebellum. (**A**–**C**) Representative images showing the number of apoptotic cells by TUNEL under normoxia (Nx,) (**A**), hypoxia (H-Ox), (**B**), and GH treatment (H-Ox+GH), (**C**). (**D**) Proportion of TUNEL area fractions expressed in percentage (%), in the different conditions. (**E**) Caspase-3 activity assay in percentage (%). (**F**) Lipid peroxidation assay. (**G**) Expression of iNOS mRNA by qPCR. Bars represent the mean ± SEM, n = 4 independent experiments by duplicate. Groups with different letters are significantly different by one-way ANOVA and Tukey’s post hoc test. Scale bar 20 µm.

**Figure 7 ijms-23-11546-f007:**
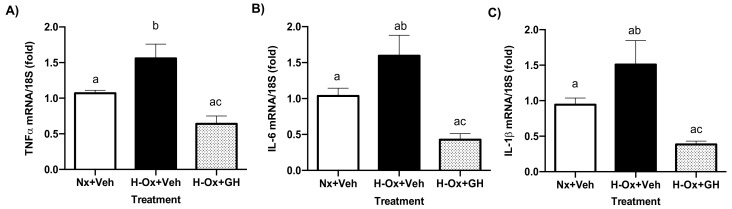
Effects of hypoxia and GH treatment upon the expression of mRNAs encoding for the proinflammatory mediators TNF𝑎, IL-6, and IL-1β in the embryonic chicken cerebellum. The relative expressions of (**A**) TNF𝑎, (**B**) IL-6, and (**C**) IL-1β mRNAs were determined by qPCR. Ribosomal 18S RNA was used as housekeeping gene control. Bars represent the mean ± SEM, n = 5 independent experiments by duplicate. Groups with different letters are significantly different by one-way ANOVA and Tukey’s post hoc test (*p* < 0.05).

**Figure 8 ijms-23-11546-f008:**
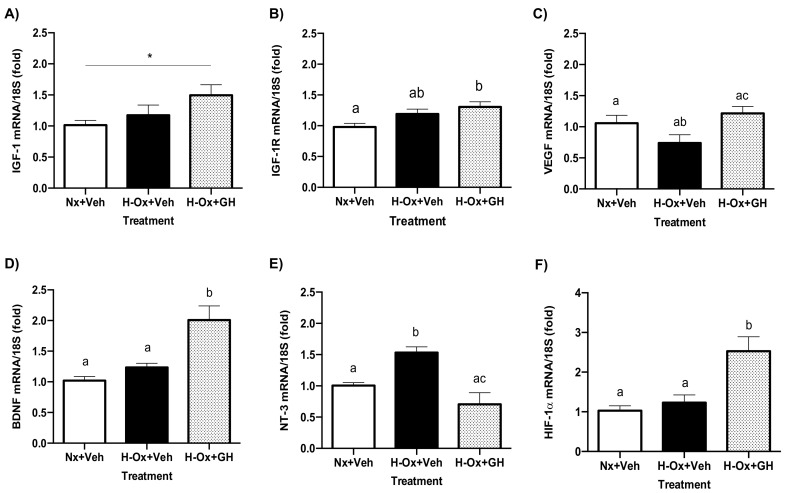
Effects of hypoxia and GH treatment upon the expression of mRNAs encoding for the neurotrophins IGF-1, VEGF, BDNF, NT-3, as well as IGF1-R and HIF-a in the embryonic chicken cerebellum. The relative expression of (**A**) IGF-1, (**B**) IGF1-R, (**C**) VEGF, (**D**) BDNF, (**E**) NT-3, and (**F**) HIF-1𝑎 mRNAs were determined by qPCR. Ribosomal 18S RNA was used as housekeeping gene control. Bars represent the mean ± SEM, n = 5 independent experiments by duplicate. Groups with different letters are significantly different by one-way ANOVA and Tukey’s post hoc test. Asterisk (*) represents differences by Student’s *t* test (*p* < 0.05).

**Figure 9 ijms-23-11546-f009:**
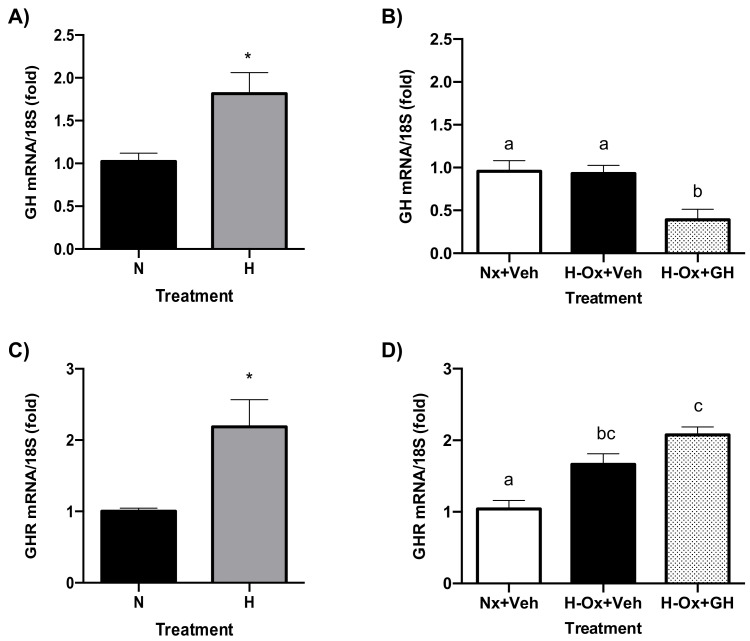
Effects of hypoxia and GH treatment upon the local expression of GH and GHR mRNAs in the embryonic chicken cerebellum. The relative expression of GH (**A**) or GHR (**C**) mRNAs was determined by qPCR after 24 h of hypoxia, or after reoxygenation for another 24 h with or without GH treatment (**B**,**D**). Ribosomal 18S RNA was used as housekeeping gene control. Bars represent the mean ± SEM, n = 5 independent experiments by duplicate. Groups with different letters are significantly different by one-way ANOVA and Tukey’s post hoc test. Asterisk (*) represents differences by Student´s *t* test (*p* < 0.01).

**Table 1 ijms-23-11546-t001:** Antibodies.

Target	Host/Type	Dilution	Source	Cat. No.
GHR	mouse/monoclonal	1:200	Thermo Fisher Scientific	MA1-82444
HIF-1α	rabbit/polyclonal	1:1000	Cell Signaling	3716S
Cy3	mouse/monoclonal	1:1000	Abcam	ab52060
GH	rabbit/polyclonal	1:10,000	LabD01	CAP2
Actin	mouse/monoclonal	1:3000	Santa Cruz Biotechnology	Sc-58673
Goat anti-Mouse IgG (H + L) Cross-Adsorbed Secondary Antibody, HRP	goat/polyclonal	1:5000	Thermo Fisher Scientific	G-21040
Goat anti-Rabbit Ig (H + L) Secondary Antibody, HRP	goat/polyclonal	1:5000	Invitrogen	656120
Rabbit anti-Mouse IgG (H + L) Secondary Antibody, FITC	rabbit/polyclonal	1:2000	Invitrogen	A16161
Goat anti-Rabbit IgG Antibody, Cy3 conjugate	goat/polyclonal	1:5000	Millipore	AP132C

**Table 2 ijms-23-11546-t002:** Oligonucleotides.

Target	Primer	Sequence (5′-3′)	Size	Accession #
GH	Fwd Rev	CGCACCTATATTCCGGAGGAC GGCAGCTCCATGTCTGACT	128 bp	NM_204359
GHR	Fwd Rev	ACTTCACCATGGACAATGCCTA GGGGTTTCTGCCATTGAAGCTC	180 bp	NM_001001293.1
IGF-1	Fwd Rev	TACCTTGGCCTGTGTTTGCT CCCTTGTGGTGTAAGCGTCT	170 bp	NM_001004384
IGF-1R	Fwd Rev	TCCAACACAACACTGAAGAATC ACCATATTCCAGCTATTGGAGC	166 bp	NM_205032.1
18S	Fwd Rev	CTCTTTCTCGATTCCGTGGGT TTAGCATGCCAGAGTCTCGT	100 bp	M59389
BDNF	Fwd Rev	AGCAGTCAAGTGCCTTTGGA TCCGCTGCTGTTACCCACTCG	167 bp	NM_001031616
VEGF	Fwd Rev	CAATTGAGACCCTGGTGGAC TCTCATCAGAGGCACACAGG	85 pb	NM_205042.2
NT-3	Fwd Rev	AGGCAGCAGAGACGCTACAAC AGCACAGTTACCTGGTGTCCT	248 bp	NM_001109762
HIF-1𝑎	Fwd Rev	GACTCCTGTTTCCACTGTAAC CAGACTCAGACACCATCAAC	153 bp	XM_040700545
TNF-𝑎	Fwd Rev	GAGCAGGGCTGACACGGAT GCACAAAAGAGCTGATGGCAG	149 bp	NM_204267.1
IL-6	Fwd Rev	AAATCCCTCCTCGCCAATCT CCCTCACGGTCTTCTCCATAAA	205 bp	NM_204628.1
IL-1β	Fwd Rev	GGATTCTGAGCACACCACAGT CAGGCGGTAGAAGATGAAGC	552 pb	XM_015297469.2
iNOS	Fwd Rev	CCAGCTGATTGGGTGTGGAT TACAGCCTTGGCCAAAATGC	194 pb	NM_204961.1

## Data Availability

The data that support finding of this study are available with the corresponding authors, upon reasonable request.

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
