# Peer review of "Growth Hormone (GH) Crosses the Blood–Brain Barrier (BBB) and Induces Neuroprotective Effects in the Embryonic Chicken Cerebellum after a Hypoxic Injury"

_ijms, 2022, doi:10.3390/ijms231911546_

Round 1
Reviewer 1 Report
The manuscript by Baltazar-Lara et al. with a title “Growth hormone (GH) crosses the blood-brain barrier (BBB) and induces neuroprotective effects in the embryonic chicken cerebellum after a hypoxic injury” examines a new aspect in the hypoxic brain injury, namely the neuroprotective effects of GH and evidence that GH crosses the BBB. The authors used an interesting model of hypoxic brain injury using chicken embryos. The authors demonstrate in a series of elegant experiments that GH injected into the chorio-allantoid vein can crossed the BBB and act in the brain by binding to its receptor. Interestingly, more GH was detected in the hypoxic than in the normoxic brain, consistent with literature showing an increase in BBB permeability under hypoxia. The abstract summarizes the presented results precisely, the introduction gives a well-structured overview of the topic and the results are clearly presented and described. The conclusion is supported by the results. The Material and Methods section is very well written and contains all the necessary details needed to repeat the experiments. All in all, this is an interesting, thoroughly executed, ready to be published with minor modifications.
I have just a few comments:
1. The last paragraph in the introduction should be shortened as it repeats information from the abstract. Rather, the scientific question and the motivation for doing this research should be emphasized.
2. Figures: the size of the scale bar should be mentioned in the figure legend as it is not possible to read the numbers in the figure
3. It would be better to change the significance statements in diagrams similar to Figure 8 A. The different letters markings are not clear and the figure legend does not explain what they mean.
4. The aspect of a possible clinical relevance of systemic injection of GH should be discussed.
Author Response
Reviewer 1
Comments and Suggestions for Authors
The manuscript by Baltazar-Lara et al. with a title “Growth hormone (GH) crosses the blood-brain barrier (BBB) and induces neuroprotective effects in the embryonic chicken cerebellum after a hypoxic injury” examines a new aspect in the hypoxic brain injury, namely the neuroprotective effects of GH and evidence that GH crosses the BBB. The authors used an interesting model of hypoxic brain injury using chicken embryos. The authors demonstrate in a series of elegant experiments that GH injected into the chorio-allantoid vein can crossed the BBB and act in the brain by binding to its receptor. Interestingly, more GH was detected in the hypoxic than in the normoxic brain, consistent with literature showing an increase in BBB permeability under hypoxia. The abstract summarizes the presented results precisely, the introduction gives a well-structured overview of the topic and the results are clearly presented and described. The conclusion is supported by the results. The Material and Methods section is very well written and contains all the necessary details needed to repeat the experiments. All in all, this is an interesting, thoroughly executed, ready to be published with minor modifications.
We express our gratitude to Reviewer 1 for the positive and supportive opinion about our study and the results we report in this manuscript. We also very much appreciate his/her recommendation to publish after minor modifications
I have just a few comments:
1. The last paragraph in the introduction should be shortened as it repeats information from the abstract. Rather, the scientific question and the motivation for doing this research should be emphasized.
Thanks for your comment. We modified the paragraph accordingly, and now it reads as follows:
Thus, in this study, we investigated if Cy3-labeled GH can cross the BBB and reach several areas in the brain, both under normoxic and hypoxic injury conditions. It was found that the fluorescent signal was distributed in several brain regions, although more intensely under the hypoxic condition. A similar distribution pattern between labeled GH and GHR in the same areas support the existence of a GH-GHR translocation mechanism, and the co-localization of Cy3-GH and GHR in the cerebellum strata indicates that GH exerts its effects directly. Also, we analyzed some of the processes implicated in the neuroprotective role of GH in the embryonic chicken cerebellum exposed to HI. It was found that GH treatment diminished brain damage in hypoxia-injured embryos through mechanisms that involve the inhibition of apoptosis, reduction of oxidative stress, and regulation of neurotrophic and inflammatory mediators. Moreover, our results showed that GH protected the injured cerebellum strata by increasing the Purkinje and granular cell survival. This work provides further evidence about the potential use of GH as a neuroprotective and regenerative treatment in perinatal asphyxia.
2. Figures: the size of the scale bar should be mentioned in the figure legend as it is not possible to read the numbers in the figure.
Thank you for pointing this out. Now, we have included the size of scale bar in all corresponding figure legends as requested, to make it clearer.
3. It would be better to change the significance statements in diagrams similar to Figure 8 A. The different letters markings are not clear and the figure legend does not explain what they mean.
It is stated that groups with different letters are significantly different from each other by one-way ANOVA, p < 0.05.
4. The aspect of a possible clinical relevance of systemic injection of GH should be discussed.
Perinatal asphyxia is a major cause of mortality and disabilities in preterm and term newborns. Infants surviving neonatal hypoxic-ischemic encephalopathy may still present motor deficits, sensory or cognitive abnormalities that persist through adolescence and/or adulthood with an increased risk for permanent neurological disorders [Millar et al., 2017; Giannopoulou et al., 2018]. It is very important to develop novel therapies to attend this public health problem, and results of this study contribute to analyze the GH potential to be considered as a neuroprotective agent in this regard.
Thus, the following paragraph was added in the discussion section:
Currently, hypothermia is the most common therapy used for hypoxic-ischemic encephalopathy (HIE) in neonates; however, it is effective only within the first 6 h and in moderate impact injuries [67]. Thus, it is desirable to develop new treatment alternatives that combine physical and pharmacological approaches. In this regard, it has been shown that GH administration improves cognitive function after intermittent hypoxia or stroke in rats [68,49]. Also, in humans, stroke and traumatic brain injury (TBI) decrease the levels of IGF-1, but GH therapy improves the speed of processing and memory and decreases the severity of depression, improving the quality of life of patients [69,70]. Results in children and adults report safe profile effects of long-term therapy [69,71], although more studies are needed to analyze the effectivity of GH treatment and IGF-1 levels in perinatal asphyxia, and its safety profile in newborns. Overall, results from this work provide further evidence about the potential of GH to be considered as a neuroprotective and neuroregenerative complementary therapy for HI encephalopathy.
In consequence, the following References were added to the text:
Millar, L.J.; Shi, L.; Hoerder-Suabedissen, A.; Molnár, Z. Neonatal hypoxia ischaemia: mechanisms, models, and therapeutic challenges. Frontiers in cellular neuroscience. 2017; 11, 78. DOI: 10.3389/fncel.2017.00078.
Giannopoulou, I.; Pagida, M.A.; Briana, D. D.; Panayotacopoulou, M. T. Perinatal hypoxia as a risk factor for psychopathology later in life: the role of dopamine and neurotrophins. Hormones. 2018; 17(1), 25-32. DOI: 10.1007/s42000-018-0007-7.
Roumes, H.; Dumont, U.; Sanchez, S.; Mazuel, L.; Blanc, J.; Raffard, G.; Chateil J.F.; Pellerin L.; Bouzier-Sore, A. K. Neuroprotective role of lactate in rat neonatal hypoxia-ischemia. Journal of Cerebral Blood Flow & Metabolism. 2021; 41(2), 342-358. DOI: 10.1177/0271678X20908355.
Ong, L.K.; Chow, W.Z.; TeBay, C.; Kluge, M.; Pietrogrande, G.; Zalewska, K.; Crock P.; Aberg D; Bivard A; Johnson S.J.; Walker F.R.; Nilsson M.; Isgaard, J. Growth hormone improves cognitive function after experimental stroke. Stroke. 2018; 49(5), 1257-1266. doi: 10.1161/STROKEAHA.117.020557.
Szarka, N.; Szellar, D.; Kiss, S.; Farkas, N.; Szakacs, Z.; Czigler, A.; Ungvari Z.; Hegyi P.; Buki A.; Toth, P. Effect of Growth Hormone on Neuropsychological Outcomes and Quality of Life of Patients with Traumatic Brain Injury: A Systematic Review. Journal of neurotrauma. 2021; 38(11), 1467-1483. doi: 10.1089/neu.2020.7265.
Song, J.; Park, K.; Lee, H.; Kim, M. The effect of recombinant human growth hormone therapy in patients with completed stroke: a pilot trial. Annals of Rehabilitation Medicine. 2012; 36(4), 447-457. doi: 10.5535/arm.2012.36.4.447.
Stochholm, K., & Kiess, W. Long‐term safety of growth hormone—A combined registry analysis. Clinical endocrinology. 2018; 88(4), 515-528.

Reviewer 2 Report
Manuscript ID: ijms-1911009
Manuscript title: Growth hormone (GH) crosses the blood-brain barrier (BBB) and induces neuroprotective effects in the embryonic chicken cerebellum after a hypoxic injury
Authors investigated the permeability of GH in BBB and the effects of GH against hypoxic injury in embryo chicken. The results are interesting, but there are some concerns before considering for publication.
Authors should demonstrate the novelty of this study. A lot of evidences already demonstrate the neuroprotective effects of GH against various neurological disorders.
Authors exposed to hypoxic condition in chicken embryo, but they did not demonstrate any gross morphology of embryonic brain or cerebellum. Are there any morphological findings in gross morphology? In addition, authors should demonstrate the microphotographs of cerebellum stained with H&E to give direct evidence on the changes of morphology.
Authors investigated whether GH can cross the blood-brain barrier (BBB) and reach the cerebellum. However, authors did not show any control in this experiment to use the other fusion protein with Cy3. It is crucial because authors used chicken embryo, not adult animals.
Authors observed Cy3-GH signal by western blot analysis at 26 kDa. Authors should clearly explain why the signals were detected at this molecular weight, not 22-23 kDa.
Author Response
Reviewer 2
Comments and Suggestions for Authors
Manuscript ID: ijms-1911009
Manuscript title: Growth hormone (GH) crosses the blood-brain barrier (BBB) and induces neuroprotective effects in the embryonic chicken cerebellum after a hypoxic injury
Authors investigated the permeability of GH in BBB and the effects of GH against hypoxic injury in embryo chicken. The results are interesting, but there are some concerns before considering for publication.
Thanks to Reviewer 2 for his/her comments about our manuscript and suggestions to improve it. Following are specific responses to each query raised by the referee:
1.Authors should demonstrate the novelty of this study. A lot of evidences already demonstrate the neuroprotective effects of GH against various neurological disorders.
As the Reviewer mentions, in recent years an increasing number of reports regarding neuroprotective actions of GH against several neurological injuries or disorders have appeared. However, there are few studies regarding the neuroprotective and restorative effects of GH in the developing brain during embryonic or neonatal stages. Some of the novelties contributed by this work are the following: a) we show that exogenously added GH can cross the blood-brain barrier in the embryonic brain, possibly through a GH-GHR translocation mechanism, and reaches several brain and cerebellar areas where it can exert direct actions, both under normoxic and hypoxic conditions; b) we offer histological, immunohistochemical, biochemical and molecular evidences that GH treatment is able to protect the cerebellar tissue against hypoxic injury, through several mechanisms involved in the regulation of apoptosis, lipoperoxidation, oxidative stress, neuroinflammation and the expression of several neurotrophins. These results contribute to shed light on the intimate processes where GH participates to exert its neuroprotective and regenerative effects; c) we give further evidence that exposure to neural damage stimulates the expression of local GH, as well as GHR, which suggests that it probably participates as a neurotrophin and/or growth factor in paracrine/autocrine mechanisms involved in neuroprotection; d) by working with the chicken embryo model, we demonstrate that the neuroprotective and regenerative mechanisms where GH participates are conserved among several vertebrate groups, since they are also present in birds; e) the results of this work give further evidence that GH can be considered as a potential therapeutic candidate in the treatment of hypoxic-isquemic encephalopathy derived from perinatal asphyxia.
2.Authors exposed to hypoxic condition in chicken embryo, but they did not demonstrate any gross morphology of embryonic brain or cerebellum. Are there any morphological findings in gross morphology? In addition, authors should demonstrate the microphotographs of cerebellum stained with H&E to give direct evidence on the changes of morphology.
We thank the Reviewer for this suggestion. We have now added a new Supplementary Figure 4, which contains the morphological analysis in hematoxylin-eosin stained sections, showing the disrupting effect of hypoxia injury on cerebellar tissue, as determined by disorganization of the cytoarchitecture of cerebellar layers, as well as the restorative effect of GH treatment both on the recuperation of layer thickness and granular and Purkinje cell survival. The information provided by this analysis was introduced in several parts of the text: Abstract (page 1); Incise 2.3 in the Results section (pages 6 and 7); and in the Discussion section (page 13).
3.Authors investigated whether GH can cross the blood-brain barrier (BBB) and reach the cerebellum. However, authors did not show any control in this experiment to use the other fusion protein with Cy3. It is crucial because authors used chicken embryo, not adult animals.
We appreciate the Reviewer comment about this point, and agree that it would have been a good control. We will try that in future experiments. Right now, we have no additional supply of Cy3 to conjugate to another protein and due to the pandemic restrictions have some problems to import the reagents, and they are not produced in Mexico. Thus, we are not in possibility to do the experiment now since performing it is clearly beyon the time frame given to submit the revised version of the manuscript to the Journal.
On the other hand, there are reports mentioning that in the chicken embryo the BBB can restrict the passage of macromolecules since 10-16 embryonic day of development, and is completely mature 2 days after hatching. Furthermore, in this embryonic stage, the barrier is almost structurally constituted; and functionally, it is already selective for the passage of molecules of different molecular weight. Some of these reports are included in the bibliography section:
- Wakai, S. and Hirokawa, N. Development of the blood-brain barrier to horseradish peroxidase in the chick embryo. Cell and tissue research. 1978;195: 195-203. DOI: 10.1007/BF00236719.
- Grange-Messent, V., Raison, D. and Bouchaud, C. Astrocyte-endothelial cell relationships during the establishment of the blood-brain barrier in the chick embryo. Biology of the Cell. 1996; 86:45–51.
4.Authors observed Cy3-GH signal by western blot analysis at 26 kDa. Authors should clearly explain why the signals were detected at this molecular weight, not 22-23 kDa.
This difference is due to the fact that the SDS-PAGE was run under reducing conditions (in the presence of 5% 2-mercaptoethano). It has long been described that when the 2 disulfide bonds of GH are broken the protein tends to linearize, and has a higher apparent MW (26 kDa) instead of 22 kDa (which is the real molecular mass). If, in contrast, the SDS-PAGE is run under non-reducing conditions, then the band corresponds to the 22 kDa MW.

Round 2
Reviewer 2 Report
The manuscript has been improved and I have no further comments.